# Infectious Recombinant Senecavirus A Expressing p16^INK4A^ Protein

**DOI:** 10.3390/ijms24076139

**Published:** 2023-03-24

**Authors:** Wencheng Gong, Xiaoya Zhao, Xiaoyu Tang, Long Gao, Yuan Sun, Jingyun Ma

**Affiliations:** 1Guangdong Provincial Key Lab of Agro-Animal Genomics and Molecular Breeding, College of Animal Science, South China Agricultural University, Guangzhou 510642, China; 2Guangdong Laboratory for Lingnan Modern Agriculture, Guangzhou 510642, China

**Keywords:** oncolytic, SVA-p16, recombinant virus, antitumor potential

## Abstract

Senecavirus A (SVA) is an oncolytic RNA virus, and it is the ideal oncolytic virus that can be genetically engineered for editing. However, there has not been much exploration into creating SVA viruses that carry antitumor genes to increase their oncolytic potential. The construction of SVA viruses carrying antitumor genes that enhance oncolytic potential has not been fully explored. In this study, a recombinant SVA-CH-01-2015 virus (p15A-SVA-clone) expressing the human p16^INK4A^ protein, also known as cell cycle-dependent protein kinase inhibitor 2A (CDKN2A), was successfully rescued and characterized. The recombinant virus, called SVA-p16, exhibited similar viral replication kinetics to the parent virus, was genetically stable, and demonstrated enhanced antitumor effects in Ishikawa cells. Additionally, another recombinant SVA virus carrying a reporter gene (iLOV), SVA-iLOV, was constructed and identified using the same construction method as an auxiliary validation. Collectively, this study successfully created a new recombinant virus, SVA-p16, that showed increased antitumor effects and could serve as a model for further exploring the antitumor potential of SVA as an oncolytic virus.

## 1. Introduction

In recent years, the field of genetic engineering has rapidly developed, and gene therapy has been increasingly used in tumor treatment. Despite their complexity, gene-based therapies have advantages over traditional cancer treatments such as chemotherapy and radiotherapy, including fewer side effects, better efficacy, and less pain for patients [1]. Furthermore, clinical trials have shown no deaths or serious clinical adverse events resulting from gene biologic therapy [2], highlighting the need to prioritize the development of new treatment strategies. One promising approach is the use of oncolytic viruses [3], which are viruses that can specifically target and kill tumor cells, either naturally or through genetic engineering [4]. These viruses can destroy tumor cells without harming normal cells and also activate the immune system to trigger an antitumor response [5]. Additionally, reverse genetic technology and carrying foreign genes can effectively enhance the antitumor immune response while minimizing cytokine storms in the host [6]. This approach, known as “targeted gene-virus therapy”, has a promising future in the field of oncology therapy [7].

Various oncolytic viruses are currently being studied, including adenovirus, cowpox virus, and Senecavirus A (SVA) [8], out of which SVA has many significant advantages. It is an oncolytic RNA virus belonging to the small ribonucleic acid virus family that does not cause harm to normal human cells but has strong oncolytic activity against tumor cells, such as neuroendocrine tumor cells and human retinoblastoma cells [9]. Moreover, SVA is an ideal virus for genetic engineering. It contains a conserved ribosomal jump sequence TNPG↓P between the 2A and 2B sequences of the genome [10], which researchers often used as an insertion site for a foreign gene. Using this site allows independent translation of the foreign gene and retention of the natural N-terminal or C-terminal end of the viral protein surrounding the foreign gene [11]. However, due to the small size of the single-stranded RNA that comprises the total SVA gene volume (approximately 7300 nucleotides {nt}), the optimal foreign gene size for SVA is limited to only 300 nt to 400 nt [12].

The INK4A gene encodes a 16 kDa small molecule protein called P16^INK4A^, also known as cell cycle-dependent protein kinase inhibitor 2A (CDKN2A) [13,14]. In cell cycle regulation, the P16^INK4A^ protein competitively binds to CDK4 and CDK6 and inhibits the phosphorylation of the retinoblastoma protein. This process prevents the activation of the E2F, thereby blocking cells from entering the G1 phase to the S phase [15]. P16^INK4A^ has promising future potential in tumor therapy by regulating the cell replication cycle and suppressing tumors. Its expression limits cell cycle progression and promotes cell senescence [16], which has broad prospects for tumor therapy.

SVA is a promising oncolytic virus with a broad range of research potential. Although it has been used as a tool for studying viral mechanisms by carrying marker genes, few studies have focused on constructing SVA oncolytic viral vectors with therapeutic transgenes or enhanced immunostimulatory efficacy. Therefore, the full potential of SVA as an oncolytic virus has not yet been fully explored. In this study, we constructed a recombinant oncolytic SVA vector expressing the human-derived p16 (SVA-p16) to further investigate the antitumor potential of SVA as an oncolytic virus and improve its efficiency in killing tumor cells. To preserve the native coding sequence of the viral protein surrounding the foreign gene, we inserted a stop–restart translation element called porcine teschovirus 2A peptide (T2A) on the side of the 2B product within the SVA polyprotein [17]. Meanwhile, the same method was used to construct an SVA viral vector carrying the green fluorescent protein (iLOV) for auxiliary validation [12]. The stability and in vivo antitumor effects of the new viral vector were evaluated and characterized in this current study, providing an effective model for exploring the antitumor potential of SVA as an oncolytic virus.

## 2. Result

### 2.1. Construction and Characterization of Recombinant SVA Viruses

As shown in Figure 1A, a panel of fragments that spans the entire SVA-CH-01-2015 genome was assembled into p15A to yield p15A-SVA-cDNA, which is the backbone of p15A-SVA-p16 and p15A-SVA-iLOV. In the virus infectious clone, a CMV promoter was placed at the 5′ terminus of the virus genome, and a polyA tail of 22 residues, SV40, and HDVr was inserted at the 3′ end (Figure 1A). A toxic gene fragment, ccdB, was inserted into the TNPG↓P sequence of the backbone plasmid 2A by homologous line-loop recombination, after which the intermediate transition plasmids were extracted after screening for the correct insertion in a specifically engineered bacterium. The p16 and iLOV genes, which combined with the T2A peptide, replaced the ccdB fragment by line-loop recombination. The resulting recombinant plasmids obtained were named p15A-SVA-p16 and p15A-SVA-iLOV (Figure 1B). During the translation of the recombinant SVA polyprotein, the ribosome initially skipped at the TNPG↓P of the SVA 2A protein and then skipped once again at another TNPG↓P in the T2A protein. Two consecutive events of ribosomal skipping led to releasing a recombinant protein. The remainder of the SVA-CH-01-2015 polyprotein was then continued to be translated within the framework. This strategy retains the advantage of preserving the native N- or C-terminus of the viral protein surrounding the foreign gene.

### 2.2. Rescue and Characterization of Recombinant SVA Viruses

To rescue the recombinant viruses, not I-linearized p15A-SVA-cDNA, p15A-SVA-p16, and p15A-SVA-iLOV were separately purified to transfect into BHK-21 cells. After 72 h post-transfection (hpt), the plasmid-transfected cells were subjected to 2 freeze-and-thaw cycles and passaged on BHK-21 and ST-R cells for 15 consecutive cycles. The rescued viruses SVA-cDNA, SVA-p16, and SVA-iLOV exhibited similar cytopathic effects like those of the parental virus, causing cell rounding and shedding (Figure 2A). Moreover, fluorescence microscopy analysis revealed the green fluorescent signals of iLOV (green) in BHK-21 cells infected with SVA-iLOV, and the signals were consistently observed during passaging (Figure 2B). Western blot analysis of samples collected from BHK-21 cells and Ishikawa cells infected with SVA-p16 revealed clear detection of p16-T2A bonds (19 kDA) (Figure 2D,E). Confocal microscopic further confirmed the successful expression of the foreign protein p16INK4A in BHK-21 cells infected with SVA-p16, as evidenced by the green fluorescence production (Figure 2C). Blank controls showed no specific fluorescence production. The results collectively demonstrated the successful rescue and stable expression of the foreign protein p16INK4A and reporter protein iLOV in BHK-21 and Ishikawa cells.

### 2.3. Genetic Stability of the Recombinant Viruses in Seral of Passages

After transfecting the recombinant viruses into BHK-21 cells, 15 passages (P15) were made in succession and stocked with viruses from P1 to P15. During the passaging period, cytopathic effects were observed, while fluorescence was observed in the reporter group. RT-PCR was performed to analyze the SVA-p16 virus at P3, P6, P10, and P15 generations, and further sequencing was conducted for analyzing the inserted fragments. The SVA-p16 and SVA-iLOV viruses caused stable cytopathic effects during the passages, and the SVA-iLOV infected group showed stable green fluorescence activity in each generation, similar to Figure 2B. Based on the results of SVA-iLOV, it can be predicted that SVA-p16 is also genetically stable. The RT-PCR analysis of p16-T2A (530 bp) showed the predicted bands, indicating that the p16-T2A gene was present in all 15 generations (Figure 3A). Sequencing results also showed that there were no mutations or deletions in the foreign fragment, as shown in Figure 3B. All these results confirmed that the viral vector could maintain stable inheritance.

### 2.4. Growth Kinetics

To further investigate the growth characteristics of SVA-p16 and SVA-iLOV, BHK-21 cells were infected with the P10 virus solutions of SVA-p16, SVA-iLOV, and SVA-cDNA (control group) at an MOI of 1.0. The TCID_50_ was measured on ST-R cells in 96-well plates, and virus growth curves were also analyzed. The results showed that viruses recombinant with foreign genes had similar replication ability and proliferation characteristics to the backbone one, with all reaching the highest titer at 18 hpi. However, at 60 hpi, the viral titer started to decrease (Figure 4A,B).

### 2.5. Oncolytic Effects of SVA-p16 in Ishikawa Cells

To further investigate the oncolytic potential of SVA-p16 compared to unmodified SVA virus, Ishikawa cells were infected with different MOIs (0.01, 0.1, 1.0, and 10.0) of parental virus and P10 virus solutions of SVA-p16, respectively. Cell viability was assessed using the CCK8 assay. The results indicated that SVA-p16 had significantly stronger antitumor effects than the unmodified SVA virus at an MOI of 10 and had similar tumorolytic properties to the unmodified SVA virus at an MOI of 10.0 (Figure 5A). However, there was no significant difference in cytopathic effects between the SVA-p16 and wild-type SVA-infected groups (Figure 5B). At 12 h, cells in all virus-inoculated groups exhibited significant swelling, rounding, and shrinking and started to lyse, forming grape-like clusters and losing their characteristic morphology. By 48 h, the cells were almost completely lysed.

## 3. Discussion

SVA infection has been an emerging disease in China since the virus was first isolated in Guangdong [18]. SVA is a lysogenic cellular RNA virus of the small ribonucleic acid virus family. It replicates through an RNA intermediate, bypassing the DNA phase and cannot integrate into the host genome [9]. SVA has numerous beneficial properties that make it an attractive oncolytic virus, such as its ability to penetrate solid tumors through intravenous injection without insertion mutation and electively target tumor cells as a self-replicating RNA virus [19,20,21]. However, the potential of SVA to carry foreign genes that enhance tumor killing is not known. Previously, most genetic engineering studies on the role of SVA as oncolytic viral vectors have been limited to the recombination with reporter genes as a tool for viral surveillance or screening and identifying receptors [11,21,22]. The virus strain SVA-CH-01-2015 isolated by our laboratory has been shown to exert antitumor effects through virus replication and subsequent cell death, which leads to the killing of tumor cells [23]. In this study, we constructed the backbone plasmid p15A-SVA-clone by taking the cDNA of SVA-CH-01-2015 with the p15A eukaryotic expression vector. We inserted the foreign gene between the 2A and 2B genes and added a testis virus 2A protease cleavage site w at its c-terminus. Finally, a recombinant virus p15A-SVA-p16 carrying p16^INK4A^ with enhanced oncolytic effect and a reporter virus p15A-SVA-iLOV with auxiliary validation effect was successfully constructed. Our results indicate that an efficient reverse genetic system was established to rescue the SVA recombinant virus. Genetic stability and growth curve results demonstrated that both SVA-iLOV and SVA-p16 exhibited similar replication and growth characteristics to the parental SVA-CH-01-2015 strain.

The majority of cancerous cells develop into tumor cells by interfering with the function of the retinoblastoma (Rb) pathway [24], resulting in uncontrolled cell proliferation and impaired apoptosis. The P16^INK4A^ protein, a cell cyclin-dependent protein kinase inhibitor protein (CKI) encoded by the INK4A gene, induces cellular senescence by upregulating Rb protein activity through the P16^INK4A^/Rb pathway [25,26]. However, in human tumors, P16^INK4A^ is in a state of inhibition due to negative feedback regulation in the P16^INK4A^/Rb pathway [27,28,29]. Therefore, on the basis of the killing of oncolytic cells themselves [30], expressing foreign p16^INK4A^ protein in tumor cells can activate the P16^INK4A^/Rb pathway and improve the efficiency of tumor suppression. In this study, the CDS region of the p16^INK4A^ gene was inserted into the SVA infectious cloned virus genome after binding to T2A, meaning that the recombinant virus SVA-p16 was constructed to directly express the p16^INK4A^ gene. Meanwhile, P16^INK4A^ protein is an ideal tumor suppressor protein to be paired with SVA due to its small molecular size. Several studies have identified the possibility that foreign genes carrying too large a length may be lost in the middle of the transmission [11,12,22], illustrating that a potential limitation of modifying SVA to carry foreign genes is the length of the inserted gene. It has been established that the length of the foreign gene is most stable at 300–400 nt [12]. Additionally, P16^INK4A^ is a small molecular protein of 16 kDa, a size that meets the requirement for gene stability. Our results show that the use of oncolytic virus SVA carrying p16 would not affect its original oncolytic effect but would also lead to a sustained high expression of p16 in tumor cells with SVA infection, resulting in the effective killing of the infected cells quite effectively. Compared with the parental strain, SVA-p16 initially exhibited stronger tumor cell cytocidal properties. SVA-p16 showed stronger and more effective antitumor efficacy in Ishikawa cells, suggesting that p16 expression has overlapping and enhanced antitumor effects with SVA.

Theoretically, the combination of SVA and P16^INK4A^ gene is expected to activate apoptosis-related pathways in multiple dimensions, including endogenous, exogenous, endoplasmic reticulum stress, P53, TNF, and other pathways [31,32]. Given the notably stronger oncolytic effect of SVA-p16 relative to the parental virus, we speculate that overexpression of the p16 gene triggered the activation of additional apoptotic pathways that overlapped with the oncolytic effect of SVA. Cancer treatment has gradually shifted from monotherapy to combination therapies [33]. The development of cancer is not simply an inactivation of a single apoptotic pathway, and targeting one node signaling pathway usually leads to the activation of alternative pathways. Stan et al. reported a bioactive compound, Withaferin A (WFA), which modulates the cell cycle and causes G2 and M phase arrest in breast cancer cells. WFA could activate multiple apoptotic pathways by modulating Bcl-2, ROS, Bax, and Bak. Additionally, combining WFA with 2-deoxyglucose (2-DG) further enhances apoptosis in breast cancer cells, eliciting more effective antitumor effects than any single-target therapies. This suggests that a regimen targeting multiple apoptotic pathways may be more effective than monotherapy [34]. Although there have been many successful reports of single-targeted drugs for cancer treatment, the problems of drug resistance and tumor recurrence cannot be circumvented [35]. In this context, we speculate that in comparison to the wild-type strain, the SVA-p16 strain targets various cell apoptosis pathways, thereby eliminating tumor cells more effectively. On the other hand, the enhanced ability of the SVA-p16 virus strain to lyse tumor cells might be attributed to its increased replication efficiency in Ishikawa cells, with a greater number of virus particles contributing to a more effective lytic impact. These speculations will be tested in future experiments using techniques such as omics analysis, quantitative PCR, Western blot, and virus titer determination. To fully understand the molecular mechanism underlying SVA-p16’s stronger antitumor properties, future studies will involve transcriptome sequencing analysis and other molecular biology techniques.

In summary, this study mainly successfully constructed an SVA recombinant vector carrying an antitumor gene P16^INK4A^. The vector was shown to be relatively stable in successive transmissions and exhibited similar cytotoxic effects to the parental virus. SVA-p16 efficiently expressed P16^INK4A^ protein in BHK-21 and Ishikawa cells and showed more potent oncolytic effects compared to the parental virus. Future studies will focus on exploring the killing efficiency of SVA-p16 on multiple tumor cells and establishing a tumor mouse model to evaluate its antitumor ability. Overall, this present study provides an effective model for exploring the antitumor potential of SVA as an oncolytic virus, which has important scientific research and application value in improving the efficacy of SVA in killing tumor cells.

## 4. Materials and Methods

### 4.1. Cell and Virus

Baby Hamster Yyrian Kidney (BHK-21) and Swine testis (ST) cells were preserved in the Key Laboratory of Animal Health Aquaculture and Environmental Control, South China Agricultural University. BHK-21 and ST cells were cultured in the DMEM (Gibco, ThermoFisher Biotechnology Products Co., Ltd., Bejing, China) supplemented with 10% fetal bovine serum (Gibco) at 37 °C and 5% CO_2_. Ishikawa cells saved in our experiment were cultured in the DMEM/F12 (Gibco) supplemented with 10% fetal bovine serum (Gibco) and Insulin (YEASEN Biotechnology Co., Ltd., Shanghai, China; 1 mg/mL, 1:1000) at 37 °C and 5% CO_2_. The wild-type strain SVA-CH-01-2015 (GenBank accession number KT321458), isolated from sows with vesicular symptoms [12], was kept at the Key Laboratory of Animal Health Aquaculture and Environmental Control, South China Agricultural University.

### 4.2. Construction of SVA Full-Length cDNA and Recombinant SVA Viruses

RNA from the strain SVA-CH-01-2015 was extracted, and cDNA was obtained by the reverse transcription as described previously [12]. Full-length amplification was performed into four separate fragments with homologous arms (named A to D). The final complement of 22 residues was added to the polyA terminus. All the above were cloned seamlessly with the p15A vector with CMV promoter, as shown in Figure 1A, and the full-length cDNA clone of SVA-CH-01-2015 was obtained and named p15A-SVA-cDNA.

To construct a recombinant SVA expressing p16^INK4A^, the combination of the porcine teschovirus 2A-like (T2A) and the stop codon-free p16^INK4A^ CDS (GenBank accession number AB060808.1) was designed and synthesized (by the GENEWIZ company, Gguangzhou, China). Then, a p16-T2A fragment with homologous arms was amplified from the synthetic gene and inserted between the 2A and 2B sequences of the p15A-SVA-cDNA. These recombinant plasmids were sequenced and designated as p15A-SVA-p16 (Figure 1B). Using the same method as above, the recombinant SVA expressing the reporter proteins was constructed and named as p15A-SVA-iLOV.

### 4.3. Recovery of Recombinant Viruses

BHK-21 cells were seeded in 12-well plates and transfected with 2.0 μg of the purified backbone plasmid p15A-SVA-cDNA and its recombinant plasmids p15A-SVA-p16 and p15A-SVA-iLOV, respectively. Transfection was performed using Lipofectamine^®^ 3000 according to the manufacturer’s instructions. After 72 h of transfection, the cell culture supernatant of BHK-21 cells was transferred to BHK-21 cells and ST-R cells, respectively. Cytopathological effects (CPE) were monitored daily after infection. The recombinant virus recovered from p15A-SVA-cDNA was named SVA-cDNA, and SVA-p16 and SVA-iLOV were rescued with the same method.

### 4.4. RT-PCR Analysis

The culture supernatant from the BHK-21cells of rSVA-p16 at passage 3, 6, 10, and 15 (P3, 6, 10, and 15) was harvested for extracting viral RNA by a Viral RNA/DNA Extraction Kit (Takara, Dalian, China). The viral RNAs of P3, P6, P10, and P15 virus stocks were extracted and reverse transcribed into cDNA. The extracted RNA was used as a template for RT-PCR analysis using a High Fidelity One-Step RT-PCR Kit (Takara, Dalian, China). The forward/reverse primers, F1/R1 (Table 1), were designed for amplifying a 530-bp fragment. The RT-PCR reaction underwent 50 °C for 30 min, 94 °C for 2 min and then 30 cycles at 98 °C (10 s), 55 °C (5 s), and 72 °C (35 s). The plasmids p15A-SVA-p16, cDNA of wild-type SVA (wt-SVA-cDNA) and ddH2O were simultaneously subjected to PCR analysis as control groups using the F1/R1 primer pair. The PCR reaction contained 2 × PrimeSTAR Max Premix (Takara, Dalian, China) and underwent 30 cycles at 98 °C (10 s), 55 °C (5 s), and 72 °C (10 s). RT-PCR and PCR products were detected by agarose gel electrophoresis, followed by Sanger sequencing for analyzing the RT-PCR product. SVA-iLOV only needs to be detected by fluorescence observation.

### 4.5. The Growth Curve of Recombinant Viruses

Monolayers of BHK-21 cells were grown in 25 cm^2^ cell culture bottles and then infected in triplicate with P10 recombinant viruses at a multiplicity of infection (MOI) of 0.1. After incubation at 37 °C for 1 h, the medium was removed and then cultured in DMEM containing 2% FBS and 1% antibiotic. The supernatants were collected at 6, 12, 18, 24, 36, 48, and 60 h post-infection (hpi) and then were titrated in ST-R cells with a 50% tissue culture infective dose (TCID50) per 100 μL strategy, according to the Reed and Muench method [36].

### 4.6. Western Blot Analysis

To detect the foreign gene protein expression, BHK-21 cells spread in a single layer of growth in 6-well plates were infected (MOI = 1.0) with the P6 and P10 virus solutions of rSVA-p16 for 24 h. The protein samples were collected 48 h after infection. After centrifugation at 15,000× *g* for 15 min, cell debris was removed. The cell lysate was mixed with 4 × Laemmli loading buffer (Bio-Rad, Hercules, CA, USA) and denatured at 95 °C for 6 min. Proteins were separated on a 12.5% SDS-PAGE gel and blotted onto a nitrocellulose membrane. The membrane was blocked with 5% skim milk in 1 × phosphate-buffered saline (PBS) at 4 °C overnight. To detect the expression of p16^INK4A^ and glyceraldehyde-3-phosphate dehydrogenase (GAPDH), the membrane was incubated separately with primary antibody, p16^INK4A^ rabbit polyclonal antibody, and GAPDH mouse monoclonal antibody, respectively, at room temperature. After 1 h of incubation, the membrane was washed with PBST 3 times and then probed with secondary antibodies, Goat anti-Rabbit IgG (HRP) and Goat anti-Mouse IgG (HRP). The membrane was incubated at room temperature for 1 h, and target proteins were visualized using the Automatic digital gel image analysis system (Tanon 2500). The same method and procedure were used to infect Ishikawa cells with the P10 and P15 virus solutions of rSVA-p16 to detect the expression of recombinant virus in tumor cells. At last, the target proteins were visualized using the chemiluminescence imaging system.

### 4.7. Indirect Immunofluorescence Assay (IFA)

BHK-21 cells were infected (MOI = 1.0) with the P10 virus solutions of SVA-p16 for 24 h while setting up a blank control and then fixed with 4% paraformaldehyde for 30 min. After fixation, cells were washed 3 times with PBS and then permeated with 0.1% Triton X-100 for 30 min. After permeation, cells were washed 3 times with PBS and blocked in skim milk with 5% concentration at 37 °C for 1 h. Subsequently, cells were incubated with the anti-p16 polyclonal antibody at 37 °C for 2 h, washed 3 times with PBS, and incubated with the Alexa Fluor^®^ 488 conjugate (Thermo Fisher, Waltham, MA, USA) at 37 °C for 1 h in a darkroom. Cells were washed three times with PBS, coated with DAPI, and visualized under the fluorescence microscope.

### 4.8. Oncolytic Effects of SVA-p16 in Ishikawa Cells

Ishikawa cells by in vitro. Ishikawa cells growing in 96-well plates were infected with the P10 virus solutions of SVA-p16 and SVA-CH-01-2015, separately, at the MOI of 0.01, 0.1, 1.0, and 10.0. Each group was set up with 6 replicates, and each replicate group had 8 wells. The control group was incubated in a complete DMEM-F12 medium. After incubation for 48 h at 37 °C in a 5% CO_2_ incubator, the absorbance (OD) at 450 nm wavelength was measured in an enzyme marker using the detection reagent of cytotoxicity assay (CCK8) to determine the cell viability.

### 4.9. Statistical Analysis

Statistical analyses were performed through the one-way analysis of variance (ANOVA) followed by Tukey’s post hoc test using GraphPad InStat Prism software (version 5.0) to establish variations between the indicated groups. *p*-values less than 0.05, 0.01, or 0.001 were assessed for statistical significance. The results were finally presented in graphical form.

## Figures and Tables

**Figure 1 ijms-24-06139-f001:**
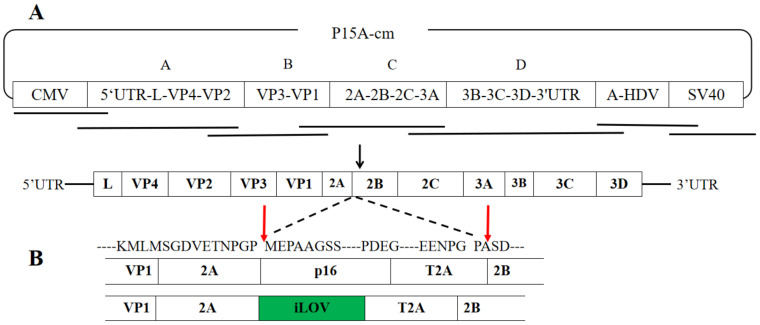
The schematic diagram for the construction of SVA. (**A**) Schematic diagram of the full-length SVA-CH-01-2015 genome and construction of the full-length cDNA clone. (**B**) Schematic presentation of the insertion of p16^INK4A^ and iLOV into SVA-CH-01-2015 genome. Cleavage of polypeptide by ribosome skipping was indicated by arrowhead. Green fluorescent of iLOV gene was indicated by green color in the figure.

**Figure 2 ijms-24-06139-f002:**
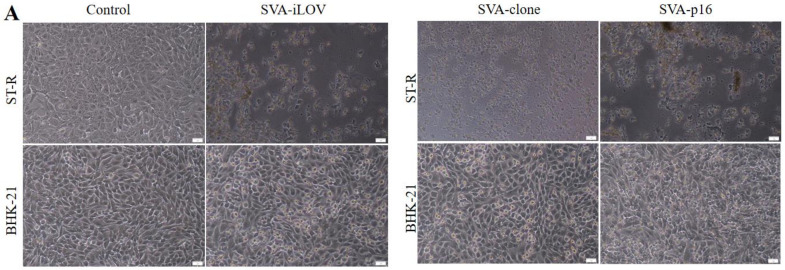
Recovery and characterization of recombinant virus. (**A**) ST-R and BHK-21 cells were infected, respectively, with SVA-clone, SVA-p16, and SVA-iLOV. After 24 h, CPE was observed under microscopy (10×). The ruler: 5 μm. (**B**) The fluorescence was observed in the SVA-iLOV-infected BHK-21 cells at 18 h (10×). The ruler: 100 μm. (**C**) Indirect immunofluorescence assay (IFA) of SVA-p16 and non-infected-BHK-21 cell monolayers (MOI = 1.0) at 18 h using anti-p16^INK4A^ polyclonal antibody (10×). The ruler: 100 μm. (**D**,**E**) Cell lysates from the recombinant viruses infected BHK-21 and Ishikawa cells, respectively, with PN-SVA-p16, and non-infected cells were harvested at 36 h. The expression of recombinant proteins was analyzed by Western blot using p16 polyclonal antibody.

**Figure 3 ijms-24-06139-f003:**
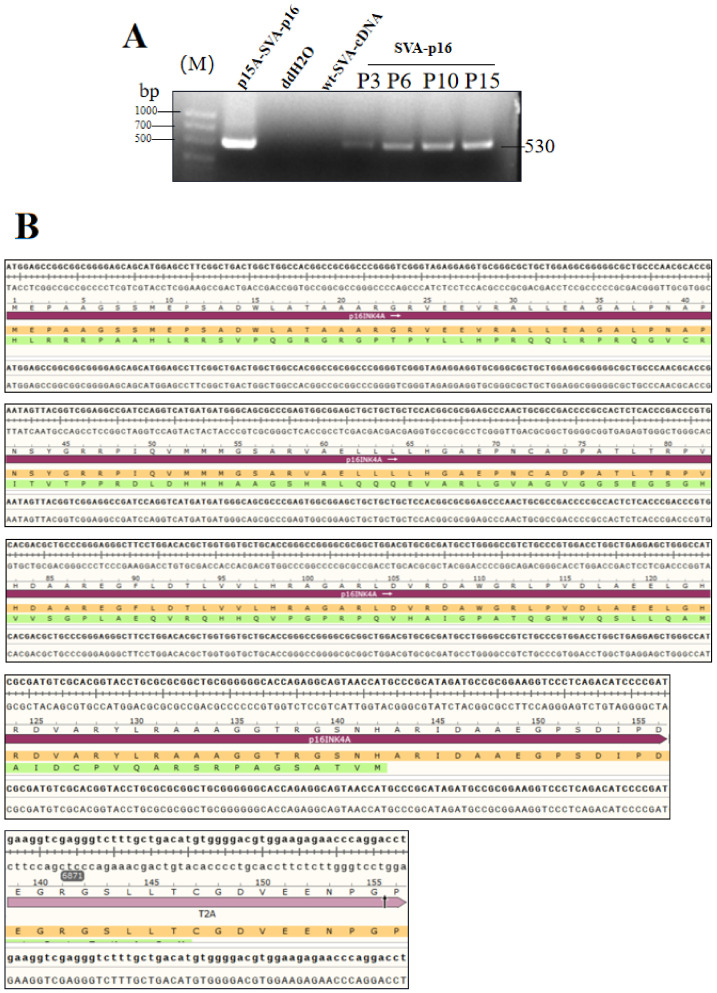
Genetic stability and growth curve of SVA recombinant viruses. (**A**) PCR amplification of the p16 gene in the SVA-P16 of P3, P6, P10, and P15 virus stocks with the p15A-SVA-p16 and wild-type SVA as controls. (**B**) Sequencing analysis of the p16 and T2A.

**Figure 4 ijms-24-06139-f004:**
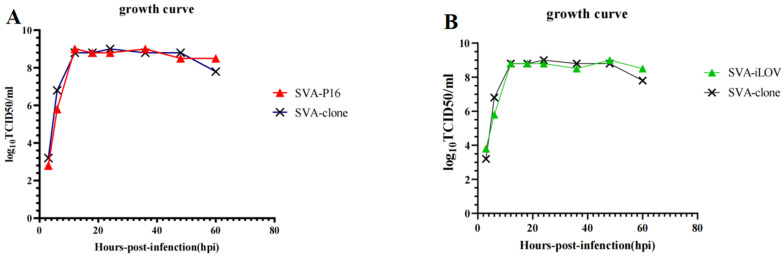
Growth curve of SVA recombinant viruses. (**A**,**B**) ST-R cells were infected with SVA-p16 and SVA-iLOV at the MOI of 0.1, and the virus titers in the cell culture supernatants were analyzed at 6, 12, 18, 24, 36, 48, and 60 hpi.

**Figure 5 ijms-24-06139-f005:**
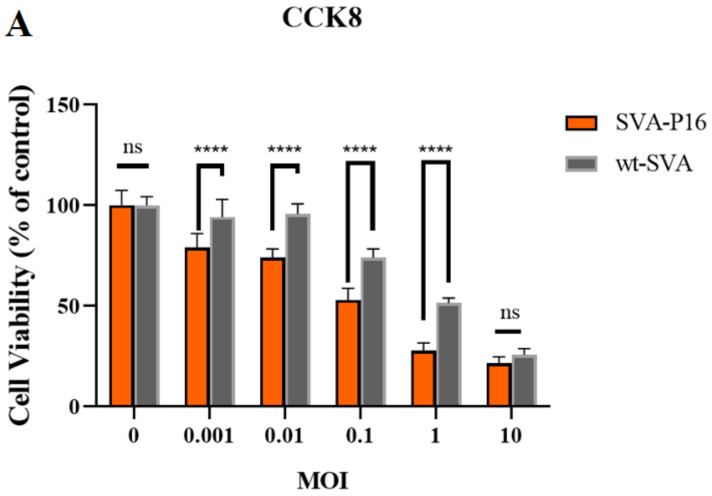
Oncolytic effects in Ishikawa cells. (**A**) Ishikawa cells were infected, respectively, with SVA-p16 and wild-type SVA as controls at the MOI of 0.01, 0.1, 1.0, and 10.0. Each bar represents the average of six independent parallels with standard deviation. Statistically significant differences with wild-type SVA are indicated (ns > 0.9999; **** *p* < 0.0001). (**B**) Ishikawa cells were infected, respectively, with wild-type SVA and SVA-p16 at the MOI of 1.0 after 24 h (10×). The ruler: 5 μm.

**Table 1 ijms-24-06139-t001:** The primers of RT-PCR.

Primers	Sequences (5′to 3′)	Length of RT-PCR Product (bp)		
		rSVA-p16	p15A-SVA-p16	rSVA
F1	ctgatgcaatcaggcgacgt	530	530	None
R1	gtcagcaaagaccctcgacc			

## Data Availability

The data presented in this study are available upon request from the corresponding author.

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
