# Peer review of "Infectious Recombinant Senecavirus A Expressing p16INK4A Protein"

_ijms, 2023, doi:10.3390/ijms24076139_

Round 1
Reviewer 1 Report
Overall this manuscript is interesting and the presented data appears to support the conclusions drawn. Major improvements in the overall presentation of the manuscript are required before it would be considered suitable for publication. There are many typographical errors (missing spaces, incorrectly formatted references, incomplete sentences, etc) that could have been identified and fixed with even a cursory proofread of the manuscript. Far too many to list in detail.
I have made some specific comments below for the authors to consider.
The figures with photographs of cells should at the very least include the magnifications used or a scale bar.
Line 9 suggest revision “Senecavirus A (SVA)”
Line 10 The sentence ending in “engineered for editing” seems incomplete. Please review.
Line 11 suggest revision “virus”
Line 14 suggest revision “replication kinetics to the parent virus and was genetically stable”
Line 18 suggest revision “study, that could provide”
Line 42 consider replacing “pathogenic” with “cytolytic”
Pathogenic refers to the capacity to cause disease, whereas in the context of this paper, cytolytic may be more appropriate.
Line 52 The beginning of the sentence appears to be missing, please review and modify it as appropriate.
Line 71 the authors state: “inserted porcine testicular virus 2A-like (T2A)”, is this correct?
I have not heard of this virus, however the “2A” peptides are usually associated with picornaviruses from the teschovirus genus. Perhaps the authors are referring to porcine teschovirus 1?
Line 92 suggest revision “to releasing a recombinant protein”
The authors should carefully review this part of the manuscript to ensure that they are not switching between molecules, namely DNA, RNA protein, in the production, expression and translation of their protein of interest.
Line 120 Fig 2B What is shown in the BHK-21 control panel? There is no definition/resolution in this image. Similarly, the DAPI panel appears to be out of focus.
Line 122 Fig 2D and Fig 2E – The authors should revise these figures so that the images shown do not overlap with the added text. Ideally, the wells of the gels would be numbered, and the explanations provided in the figure legend.
I would suggest the authors supply the full Western blot images as supplemental files.
Line 126 There does not appear to be a scale bar on Fig 2B.
Line 146 There does not appear to be a growth curve shown in this figure.
Line 146 suggest replacing “identification” with “PCR amplification of the p16”
Line 148 It is not readily apparent what Fig 3B is supposed to illustrate.
Line 181 Should “empty expression vector” be “wild type SVA”?
Also Fig 5A suggests that all significant differences were detected at the p<0.0001 level (****), if this is correct, then there is no need to have the * p<0.05; ** p<0.01; *** p<0.001 in the legend.
Was the level of significance p<0.0001 at MOI=0.001?
Line 184 – There seems to be numerous (and obvious) errors in the reference formatting in this section.
Line 233 Is the CDKN2A gene really an oncogene? This seems to contradict the stated function of inducing cellular senescence (line 211).
Reviewer 2 Report
The authors constructed SVA-CH-01-2015 expressing p16INK4A (SVA-p16) and showed replication similar to the parental virus but stronger anti-tumor effect on Ishikawa cells. Therefore, SVA-16p was found to be an effective novel oncolytic agent.
Systemic administration of oncolytic viruses is a promising route to deliver viruses to a wide range of primary and metastatic lesions. Therefore, avirulent SVA and its modified viruses may be useful to improve efficacy in multiple malignant tumor lesions. However, there are some points to reconsider, as described below.
The authors stated that “further studies will contribute to explore both the killing of multiple tumor cells”. So far, SVA has been tested its antitumor ability against human tumors such as small cell lung cancer, neuroblastoma, Ewing sarcoma, and rabdomyosarcoma. The selective oncolytic activity of SVA is determined by its affinity for a specific cellular receptor, the TEM8. In this study, the authors used Ishikawa cells as a human tumor cell line. The reason for using this cell line, its origin, and the level of expression are important. Other types of cell lines with different levels (high, low) of the TEM8 should be investigated.
The authors intended to enhance the cytotoxicity of SVA by introducing the p16 gene. Expression of p16 in SVA-p16-infected cells was confirmed (Figure 2), and the anti- tumor capability of SVA-p16 against Ishikawa cells was found to be greater than that of wt-SVA at low input MOI (Figure 5). However, it is not unclear how p16 contributes to the antitumor ability of SVA-p16. Did the over-expressed p16 act as an inhibitor of CDK4/6 and cause cell cycle arrest/cell death in infected cells?
Lines 35-36: “without causing or lowering the host’s own cytokine storm” This sentence needs improvement.
Line 52: “protein” would be “P19INK4A protein”.
Line 81: “p15A-SVA-p16and” should be “p15A-SVA-p16 and”
Lines 111-113: (Figure 2C, D) and Figure 2E should be (Figure 2D, E) and Figure 2C. If not, the markings in Figure 2 should be changed.
Figure 2, lmmunoblot: ”p6-sva-p16 and p10-sva-p16” should be “P6-SVA-p16 and P10-SVA-p16”.
Line 132: “15 passages” should be changed to “15 passages (P15)”.
Figure 3: A, “p3, p6, p10, p15” should be “P3,P6, P10, P15”. B, the portion of the SVA-p16 gene that was sequenced should be indicated in the figure so that it can be seen. The passage level of the virus should be stated.
Line 152: “while” is duplicated.
Figure 5A: The graph indicates that viral spread in monolayer is faster in SVA-p16 than in wt-SVA, possibly resulting in higher virus yields in SVA-16 compared to wt-SVA. This possibility should be considered.
Lines 195 and 222: What does 111222 mean?
References should be checked. For example, ref 12 not found.
Round 2
Reviewer 2 Report
The authors respond appropriately to the points rasied by the reviewer.